# Optimizing for Generalization in Machine Learning with Cross-Validation Gradients

## Abstract

Cross-validation is the workhorse of modern applied statistics and machine learning, as it provides a principled framework for selecting the model that maximizes generalization performance. In this paper, we show that the cross-validation risk is differentiable with respect to the hyperparameters and training data for many common machine learning algorithms, including logistic regression, elastic-net regression, and support vector machines. Leveraging this property of differentiability, we propose a cross-validation gradient method (CVGM) for hyperparameter optimization. Our method enables efficient optimization in high-dimensional hyperparameter spaces of the cross-validation risk, the best surrogate of the true generalization ability of our learning algorithm.

## 1 Introduction

The ultimate aim of a supervised learning method is *generalization*, that is, achieving good prediction ability on unseen test data given only a finite set of training data. The generalization capability of learning algorithms should be the primary criterion for model selection, yet an algorithm's generalization capability is a somewhat elusive quantity that is challenging to optimize for. In this paper we introduce a method to optimize directly for the closest available proxy to generalization performance: cross-validation loss.

We begin with a formal description of the overall goal in predictive learning, which also serves as an introduction to notation used throughout the paper. The task of predictive learning involves deriving a prediction function from a finite set of training data. More formally, suppose that $(x, y) \in \mathcal{X} \times \mathcal{Y}$ have some joint probability distribution. We have access to a finite dataset of $N$ training examples $z_i \in Z = \mathcal{X} \times \mathcal{Y}$ drawn i.i.d. from the joint distribution, denoted

$$S = \{z_1, z_2, \ldots, z_N\}.$$

We are given (or specify ourselves) a cost function $c(\hat{y}, y) : \mathcal{Y} \times \mathcal{Y} \to \mathbf{R}_+$ that quantifies the displeasure incurred when $\hat{y}$ is predicted instead of $y$. Denoting the function space from input to outputs as $\mathcal{F} = \mathcal{Y}^{\mathcal{X}}$, we define the loss of a function on a training example $z = (x, y)$ as $l(f, z) = c(f(x), y)$. Then, given a prediction function $f \in \mathcal{F}$, we define the population risk as

$$\mathbf{E}_z[l(f, z)],$$

and the target function $f^* \in \mathcal{F}$ as the function that minimizes the population risk. The population risk represents how much loss we incur, on average, on the full joint distribution, and is the quantity we would like as small as possible. In this paper, we consider parametric prediction functions, that is, $f$ is parameterized by a vector $\theta$, denoted $f(x; \theta)$[1]. For example, in linear regression, $\mathcal{X} = \mathbf{R}^n$, $\mathcal{Y} = \mathbf{R}$, $c(\hat{y}, y) = (\hat{y} - y)^2$, and $\mathcal{F}$ is the set of all affine functions parameterized as $f(x; \theta) = \theta^T x + \theta_0$.

We are then tasked with designing a learning algorithm $\mathcal{A} : Z^N \to \mathcal{F}$, which is a function that maps a dataset $S$ to a prediction function. Without substantial knowledge of the actual joint distribution, or assumptions about the target function $f^*$, it is extremely unlikely that $\mathcal{A}$ will ever reproduce the

---

[1]Nonparametric learning algorithms exist, *e.g.*, k-nearest neighbor, but are challenging to analyze with our method.

exact target function. However, our goal is to minimize the population risk of the learning algorithm

$$R(\mathcal{A}, S) = \mathbf{E}_z \left[ l(\mathcal{A}(S), z) \right] \tag{1}$$

which is a random variable that depends on $S$, our dataset. To make this problem of searching for learning algorithms tractable, we similarly parameterize our learning algorithm $\mathcal{A}$ by a vector $\alpha \in \mathbf{R}^d$, denoted $\mathcal{A}_\alpha$. These are known as the "hyperparameters" or "meta-parameters" of the learning algorithm, and can play many important roles: they can perform regularization, enforce sparsity, or even guide feature selection (Hastie et al., 2001). The quantity we would then like to optimize is the expected population risk, or

$$L(\alpha) = \mathbf{E}_S \left[ R(\mathcal{A}_\alpha, S) \right]. \tag{2}$$

It is impossible to exactly calculate (2) with a finite dataset $S$, as there are two expectations that both involve an unknown probability distribution. What we *can* do is construct a Monte Carlo estimate of the an algorithm's expected population risk using a technique known as cross-validation. We first partition $S$ into $K$ partitions $\mathcal{T}_j, \mathcal{V}_j, j = 1, \ldots, K$ (that is, $\mathcal{T}_j \cap \mathcal{V}_j = \emptyset$ and $\mathcal{T}_j \cup \mathcal{V}_j = [n]$). Then our cross-validation risk, as a function of $\alpha$, is

$$L_{\mathrm{cv}}(\alpha) = \frac{1}{K} \sum_{j=1}^{K} \frac{1}{|\mathcal{V}_j|} \sum_{i \in \mathcal{V}_j} l(\mathcal{A}_\alpha(\mathcal{T}_j), z_i) \tag{3}$$

and is readily calculated. We first apply the algorithm to each training set and then average the loss on each corresponding validation set. The first sum in (3) corresponds to the expectation in (2), and the second sum corresponds to the expectation in (1). Setting $K = 1$ reduces to simple out-of-sample validation and an arbitrary $K$ reduces to the common $K$-fold cross-validation estimate (provided $T_j$ form a partition of $\{1, \ldots, N\}$ and $|T_j| = N - \frac{N}{K}$). Thus, this formulation can be viewed as a generalization of cross-validation. (See Kohavi & John (1995) for a longer discussion about this general framework.) In cases where the class of models to be used for learning are known, we have reduced the predictive learning problem to the problem of selecting of a hyperparameter vector $\alpha$ to minimize the cross-validation loss. Even in the simplest cases, however, the objective in (3) is nonconvex in $\alpha$, and in many cases not even continuous (*e.g.*, the $0 - 1$ classification loss), which can make optimization of this quantity tricky.

## 1.1 SUMMARY OF RESULTS

Our first result is to demonstrate that we can find $\nabla_\alpha L_{\mathrm{cv}}(\alpha)$ for many common convex machine learning algorithms (*e.g.*, logistic regression, elastic-net regression, support vector machines), provided the cross-validation loss function is differentiable (Section 3). In those algorithms, $\alpha$ often plays the role of regularizer or defines a feature map (in the case of SVM kernels). In the case where $\alpha$ is low-dimensional, (3) can be optimized by exhaustive search without incurring too much cost. However, if we want to design our machine learning algorithms with more expressive regularizations or feature maps, exhaustive search over our hyperparameter space becomes prohibitive.

Our second contribution is to propose the cross-validation gradient method (CVGM), which makes it possible to optimize cross-validation loss over high-dimensional hyperparameter spaces via gradient descent techniques (Section 4). We test the CVGM on an elastic-net regression problem to optimize two hyperparameters and on a more ambitious synthetic classification problem to optimize an entire neural network that serves as a kernel function. In this case, the parameters of the neural network are the hyperparameters (Section 5).

## 2 RELATED WORK

There have been many proposed approaches for the problem of hyperparameter optimization, which roughly fall into two camps based on whether or not they use gradients.

### 2.1 GRADIENT-FREE METHODS

**Exhaustive Search** Exhaustive search, also known as grid search, restricts the possible set of $\alpha$ to a (finite) set $\{\alpha_1, \ldots, \alpha_n\}$, usually by discretizing the parameter search space into a regular grid.

Then one exhaustively computes (3) for each $\alpha_i$ and chooses the argmin. The main disadvantage of exhaustive search is that its complexity (to find an approximate minimum) scales exponentially with the dimension $d$, making it prohibitive for practitioners to successfully apply exhaustive search to $d$ greater than 5 or 6.

**Random Search**   Random search for hyperparameters involves repeatedly specifying a probability distribution over $\mathbf{R}^d$, sampling from it, and evaluating (3). Quite unintuitively, random search can be more efficient than exhaustive search, even with a simple probability distribution. This is because, in practice, only a few of the hyperparameter dimensions matter (Bergstra & Bengio, 2012).

**Bayesian Optimization**   Bayesian regression allows us to predict a distribution over $L_{\mathrm{cv}}(\alpha)$, further allowing us to query $\alpha$ that maximize a surrogate function, *e.g.*, probability of improvement or expected improvement (Močkus, 1975; Snoek et al., 2012). The regression is usually carried out with Gaussian processes (GPs) (Rasmussen, 2004). However, random search still remains a fierce competitor to the (substantially more complicated) Bayesian optimization approach.

## 2.2 GRADIENT-BASED METHODS

**Implicit Differentiation**   Most learning algorithms $\mathcal{A}_\alpha$ are solving some parameterized optimization problem, that is, optimizing some objective function. Under certain conditions, one can apply the well-known implicit function theorem (Dontchev & Rockafellar, 2014) to the optimality conditions of the objective function, and calculate the gradients of the loss function. Larsen et al. (1998) were the first to propose this, in the context of neural networks, when the objective function includes a regularization term that is linear in the regularization parameters. Bengio (2000) further derived the gradients for a general (unconstrained and differentiable) training criterion along with an efficient way of calculating the gradient for a quadratic training criterion, and applied the algorithm to weight decays for linear regression. These results were then extended to support vector machines (SVMs) (Chapelle et al., 2002; Keerthi et al., 2007) and applied to log-linear models (Foo et al., 2008) and ridge regression (Pedregosa, 2016).

This paper seeks to generalize these methods and provide exact conditions under which $\mathcal{A}_\alpha$ is actually differentiable. In short, when $\mathcal{A}_\alpha$ is a convex optimization problem parameterized by $\alpha$, under certain conditions that are satisfied by many common learning algorithms, we can find exact cross-validation gradients.

**Iterative Differentiation**   In addition to approaches based on implicit differentiation, there are also approaches based on iterative differentiation, *i.e.*, they unroll the optimization procedure in $\mathcal{A}$ to calculate gradients. Many large-scale machine learning algorithms perform a variation of gradient descent, and since gradient descent is a sequence of analytic updates to the parameters, $\mathcal{A}_\alpha$ can be unrolled (or "reverse-mode" differentiated) with respect to $\alpha$ by recursively applying the chain rule to the updates in backwards order. Domke (2012) was the first to propose this, deriving backpropagation rules for the heavy-ball method and LBFGS. Since most large-scale machine learning problems in practice are (approximately) solved using variations of the stochastic subgradient method (also known as SGD), the "learning rate" parameter has a large impact on the convergence and training speed of nonconvex models, *e.g.*, neural networks. Maclaurin et al. (2015) extended the results of Domke to the case of stochastic gradient methods, and as a result, the authors were able to update the learning rate throughout the learning process. The advantage of these methods are that they can be applied to any large-scale machine learning problem that uses a stochastic subgradient method. The main limitations of these methods, however, are that the use of finite precision arithmetic when recursively applying the chain rule can lead to inaccuracies in the gradient calculation and that one can encounter exploding or vanishing gradients from repeated application of the chain rule.

## 3 EXACT DIFFERENTIABILITY OF LEARNING ALGORITHMS

Recent work by Barratt (2018) provided necessary and sufficient conditions for a parameterized convex optimization problem to be differentiable. We review the results here, and refer the reader to

the paper for more details. The setting is a parameterized convex optimization problem

$$\begin{aligned} \text{minimize} \quad & f_0(x, \alpha) \\ \text{subject to} \quad & f(x, \alpha) \preceq 0, \\ & h(x, \alpha) = 0. \end{aligned} \tag{4}$$

where $x \in \mathbf{R}^n$ is the optimization variable, the functions $f_0$ and $f$ are convex for fixed $\alpha$ and $h$ is affine for fixed $\alpha$. Let $s(\alpha) = (\tilde{x}, \tilde{\lambda}, \tilde{\nu})^T$ denote the optimal $x, \lambda, \nu$ for a given $\alpha$ in (4), where $\lambda$ and $\nu$ are Lagrange multipliers, *i.e.*, that satisfy the Karush-Kuhn-Tucker (KKT) conditions. Then define the vector-valued function

$$g(\tilde{x}, \tilde{\lambda}, \tilde{\nu}, \alpha) = \begin{bmatrix} \nabla_x L(\tilde{x}, \tilde{\lambda}, \tilde{\nu}, \alpha) \\ \mathbf{diag}(\tilde{\lambda}) f(\tilde{x}, \alpha) \\ h(\tilde{x}, \alpha) \end{bmatrix}. \tag{5}$$

where $L$ is the Lagrangian. The main result of the paper is that, for an optimal $z = (\tilde{x}, \tilde{\lambda}, \tilde{\nu})$,

$$\nabla_\alpha s(\alpha) = -\nabla_z g(\tilde{x}, \tilde{\lambda}, \tilde{\nu}, \alpha)^{-1} \nabla_\alpha g(\tilde{x}, \tilde{\lambda}, \tilde{\nu}, \alpha). \tag{6}$$

under the assumption that both $f_i$ and $g$ are twice differentiable in $x$ and $\alpha$, strong duality holds, and $\nabla_\alpha g(\tilde{x}, \tilde{\lambda}, \tilde{\nu}, \alpha) \in \mathcal{R}(\nabla_x g(\tilde{x}, \tilde{\lambda}, \tilde{\nu}, \alpha))$. In other words, we can get the derivative of $(x, \lambda, \nu)$ with respect to $\alpha$. We will focus on the derivative of $x$ with respect to $\alpha$ in this paper, however, it would be interesting to consider the derivative with respect to the dual variables $\lambda$ and $\nu$.

Since most parametric machine learning procedures can be expressed as parameterized convex programs that satisfy these conditions[2], we can conclude that, in many cases, $\mathcal{A}_\alpha$ is in fact differentiable. In fact, many machine learning procedures can even be expressed as quadratic programs (QPs) — quadratic objectives with affine inequality and equality constraints — and satisfy the conditions for differentiability, as shown in Amos & Kolter (2017) (assuming a positive definite quadratic). Further, if $l$ is differentiable with respect to the parameters of $f$, we can use the chain rule to find the gradient of the cross-validation loss with respect to the hyperparameters. We now present several examples of predictive learning algorithms that are in fact differentiable with respect to their hyperparameters. (Additional examples, including the support vector machine, can be found in the Supplementary Materials.)

**Example 3.1** (Logistic regression). In logistic regression, $\mathcal{X} = \mathbf{R}^n$ and $\mathcal{Y} = \{-1, 1\}$. As is standard in classification, we model $\mathbf{Prob}(x = 1)$. This probability is represented by the "sigmoid" function $p(x; \theta) = \frac{1}{1+\exp(-\theta^T x)}$ and we minimize a loss function that is proportional to the likelihood of the dataset under this model plus a regularization term

$$L(\theta, C) = \frac{1}{2}\theta^T \theta + C \sum_{i=1}^{N} \log(\exp(-y_i x_i^T \theta) + 1).$$

This (convex) optimization problem is unconstrained, so the derivative of the optimal solution with respect to the hyperparameter $C$ is just

$$\nabla_C \mathcal{A} = -\nabla_\theta^2 L(\theta, C)^{-1} \nabla_C [\nabla_\theta L(\theta, C)].$$

Letting $\pi_i = \frac{1}{1+\exp(-y_i \theta^T x_i)}$, the gradient is

$$\nabla_\theta L(\theta, C) = \theta + C \sum_{i=1}^{N} (\pi_i - 1) y_i x_i$$

and the Hessian is

$$\nabla_\theta^2 L(\theta, C) = I + C X^T D X$$

where $D$ is a diagonal matrix with $D_{ii} = \pi_i(1 - \pi_i)$ and the rows of $X$ are $x_i$, and is guaranteed to be positive definite. The righthand side is just

$$\nabla_C \nabla_\theta f(\theta, C) = \sum_{i=1}^{N} (\pi_i - 1) y_i x_i.$$

---

[2]Two notable exceptions to this are neural networks and decision trees, which both have nonconvex training criterions and thus one cannot guarantee finding a global minimum.

---

**Algorithm 1** CVGM

---

**Require:** $\alpha_0$: Initial hyperparameter vector
**Require:** $K$: Number of partitions
**Require:** $p$: Fraction of samples in training set
 1: Sample $K$ partitions $\mathcal{T}_j, \mathcal{V}_j, j = 1, \dots, K$ uniformly at random, where $|\mathcal{T}_j| = \lfloor pN \rfloor$
 2: **while** $\alpha_k$ not converged **do**
 3:     Run the algorithm $\mathcal{A}_{\alpha_k}(\mathcal{T}_j)$ separately on each training set
 4:     Calculate the gradient

$$g \leftarrow \nabla_\alpha \left[ \frac{1}{K} \sum_{j=1}^{K} \frac{1}{|\mathcal{V}_j|} \sum_{i \in \mathcal{V}_j} l(\mathcal{A}_{\alpha_k}(\mathcal{T}_j), z_i) \right]$$

 5:     Update $\alpha_{k+1}$ using a gradient method with the gradient $g$
 6:     Project $\alpha_{k+1}$ onto the constraint set
 7: **end while**

---

We can also take the derivative with respect to the training examples $x_i$ (or $y_i$ with a similar derivation) using the fact that

$$\nabla_{x_i} \nabla_\theta f(\theta, C) = C(\pi_i - 1) y_i I.$$

**Example 3.2** (Elastic-net regression). In regression, $\mathcal{X} = \mathbf{R}^n$, $\mathcal{Y} = \mathbf{R}$, and $c(\hat{y}, y) = (\hat{y} - y)^2$. The function $f(x; \theta) = \theta^T x$, where the intercept term is omitted for illustration. Let the $i$th row of the data matrix $X$ be equal to $x_i$ and the $i$th entry of the vector $y$ be equal to $y_i$. Elastic-net regression generalizes ridge and LASSO regression and optimizes the squared penalty with a weighted combination of $\ell_1$ and $\ell_2$ regularizers (Zou & Hastie, 2005), or solves the optimization problem

$$\text{minimize} \quad \frac{1}{2N} \|X\theta - y\|_2^2 + \lambda_1 \|\theta\|_1 + \frac{1}{2}\lambda_2 \|\theta\|_2^2. \tag{7}$$

The objective is convex, but not differentiable. To transform this into a differentiable parameterized convex optimization problem, we introduce two variables to represent the positive and negative parts of $\theta$, denoted $\theta_p$ and $\theta_n$. Then, letting $v = [\theta_p \ \theta_n]^T$, elastic-net can be expressed as the following quadratic program (QP) with $2n$ variables

$$\text{minimize} \quad \frac{1}{2}v^T \begin{bmatrix} \frac{1}{N}X^TX + \lambda_2 I & -\frac{1}{N}X^TX \\ -\frac{1}{N}X^TX & \frac{1}{N}X^TX + \lambda_2 I \end{bmatrix} v + \begin{bmatrix} -\frac{1}{N}X^Ty + \lambda_1 I \\ \frac{1}{N}X^Ty + \lambda_1 I \end{bmatrix}^T v \tag{8}$$
$$\text{subject to} \quad v \succeq 0.$$

Since we can differentiate the solution to positive-definite QPs, we can find the gradient of the optimal solution $\theta = \theta_p - \theta_n$ with respect to the hyperparameters $\lambda_1$ and $\lambda_2$, provided $\lambda_2 > 0$. To the best knowledge of the authors, this is the first derivation of the gradients of the elastic-net solution with respect to elastic-net's hyperparameters.

## 4 CROSS-VALIDATION GRADIENT METHOD (CVGM)

Building off our findings that the solution to many parametric machine learning procedures are differentiable with respect to their hyperparameters, we can now design an algorithm to minimize (3).

The algorithm is summarized in Algorithm 1. The algorithm essentially performs projected gradient descent on (3), restricting $\alpha$ to a pre-defined constraint set. It runs the learning algorithm on the training part of each cross-validation split (line 3), then calculates the loss on the held-out part of each cross-validation split, and then uses the chain rule to calculate their gradients, which are then averaged (line 4). This averaged gradient is then used to update $\alpha$ in a first-order gradient method (line 5), and then $\alpha$ is projected back onto the constraint set (line 6). Once we run the CVGM to find $\alpha^*$, we then run the learning algorithm on the full dataset to find the final prediction function $\mathcal{A}_{\alpha^*}(S)$.

There are several advantages to this method. First, it directly optimizes the quantity of interest using a gradient-based method, which can be much faster than exhaustive search. Second, if one is smart with their implementation, computing the gradient in line 4 of the algorithm costs little on top of evaluating the function $L_{\text{cv}}(\alpha)$ itself. (See Barratt (2018) and Amos & Kolter (2017) for a discussion of this.) Third, our method plays well with parallel computation. The majority of computation time is spent finding the gradient in line 4 of the algorithm. Since the gradient operation is linear, we can split up $K$ runs of the learning algorithm on the $K$ datasets over $K$ processors or compute nodes and then average the resulting gradients. Also, the algorithm can be run in parallel with different random initializations of $\alpha$ to find multiple hyperparameter settings.

There are several immediate improvements that can be made to the CVGM as stated. One improvement would be to make the sampling of cross-validation splits uniform, that is, each index appears an equal number of times in all of the $\mathcal{V}_j$ and $\mathcal{V}_j$. This ensures that each data point shows up an equal number of times in the cross-validation loss. Another improvement would be to use a more sophisticated optimization method, *e.g.*, accelerated or adaptive methods, but in our experiments we just use a gradient method with constant step size and found that it works quite well.

Our method requires two parameters: the number of partitions $K$ (the batch size), and the fraction of samples to include in the training set partition $p$. We expect that a value of $K$ between 16 and 128 and $p > \frac{1}{2}$ should work well in almost all scenarios. A larger $K$ leads to reduced variance, and a larger $p$ leads to a reduced number of examples held out for validation.

## 5 NUMERICAL EXPERIMENTS

We evaluate our method on synthetic regression and classification data, noting that further in-depth comparison on real datasets is needed in future work. One benefit of small synthetic experiments is that the true population risk is readily calculated, and also we can evaluate the performance of the method in the low-data regime. All of the code to run our experiments is freely available online[3].

### 5.1 SYNTHETIC REGRESSION DATA

First, we evaluate our method on synthetic regression data. There are $N = 30$ observations and $n = 10$ features. However, only 8 of the features have non-zero coefficients. We generate data via the following scikit-learn (Pedregosa et al., 2011) command:

```
X, y, coef = make_regression(N, n, n_informative=8, noise=100., \
  tail_strength=0., coef=True)
```

We also generate a test set of 1000 examples with the same command for evaluation. As is standard practice in machine learning, we normalize the features—that is, we normalize each feature to mean 0 and standard deviation 1 across the training set and then this same normalization is applied to the validation/test set before prediction.

We run an elastic-net regression (see Example 3.2) to learn a linear prediction function. For simplicity, the (unpenalized) intercept is learned using standard linear regression and then subtracted from $y$. For our projection step (line 6 of the algorithm), we require $\lambda_2 \geq \epsilon$, where $\epsilon = 1 \times 10^{-7}$, and $\lambda_1 \geq 0$. We use $K = 128$, $p = .95$, and a gradient descent step size of $2 \times 10^{-4}$. The method is implemented in PyTorch using the qpth library, which is a fast, batched, and differentiable QP library, making the algorithm efficient and scalable (Amos & Kolter, 2017). The authors note, however, that one could create a much faster implementation by making the solver specialized for elastic-net regression.

We compared the CVGM with exhaustive and random search, noting, however, that the hyperparameter optimization problem is in two dimensions and exhaustive/random search are likely to be quite competitive. We ran CVGM with an initial $\lambda_1 = 1 \times 10^{-2}$ and $\lambda_2 = 1 \times 10^{-4}$ for 100 steps. For exhaustive search, we did a grid search over a log scale for $\lambda_1 \in [1 \times 10^{-4}, 1 \times 10^{-1}]$ and $\lambda_2 \in [1 \times 10^{-4}, 1 \times 10^{-1}]$, and kept the hyperparameters that achieved the lowest cross-validation loss. For random search, we sampled uniformly at random in a log scale from the same variable

---

[3]`www.github.com/anonymous/crossval`

Table 1: Synthetic Regression Results.

| Name | Test Loss | Cross-Validation Steps |
|------|-----------|------------------------|
| CVGM | **1.080** | 100 |
| Exhaustive search | 1.207 | 100 |
| Random search | 1.084 | 100 |

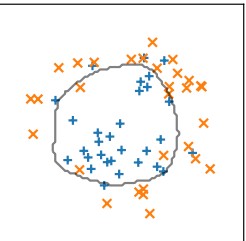

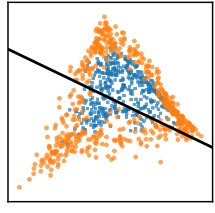 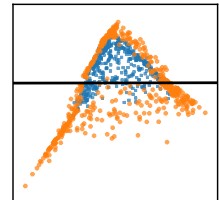 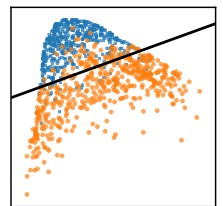 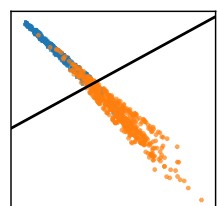

Figure 1: Top: Synthetic classification training data in two dimensions, along with the learned decision boundary (black). Bottom: Visualization of kernel applied to 1000 (unseen) test data points in iterations 1, 10, 20, and 100 from left to right, along with the linear classifier in that space (black). Note that the kernel quickly learns a manifold where the data is (approximately) linearly separable. The CVGM achieves $89.8\,\%$ test accuracy on the test set with only 60 training examples, and the Bayes (optimal) accuracy is $89.4\,\%$.

ranges as exhaustive search. The resulting final test losses (at iteration 100) of this experiment are in Table 1 and we also included a plot of the (test loss) progress of the algorithms in Figure 3 in the Supplementary Materials. CVGM ultimately achieves a test loss of $1.080$, lower than the other two methods, and the test loss (which CVGM has no access to but we do compute during training) is for the most part monotonically decreasing throughout the procedure.

## 5.2 SYNTHETIC CLASSIFICATION DATA

Next, we experiment with CVGM's ability to learn kernels from scratch on two dimensional synthetic classification data. We first generate a two dimensional dataset of $N$ examples in polar coordinates from two classes that form rings of different radii and have significant overlap. One class has the distribution $r \sim \mathcal{N}(1, .4)$ and $\theta \sim \text{Unif}(-\pi, \pi)$, and the other class has the distribution $r \sim \mathcal{N}(2, .4)$ and $\theta \sim \text{Unif}(-\pi, \pi)$. The data is then transformed into Cartesian coordinates using the transformation $(x, y) = (r \cos \theta, r \sin \theta)$. A training dataset of size $N = 60$ is displayed in the top part of Figure 1. Clearly, the Bayes decision rule for this dataset is to separate the classes at $\|x\|_2 = 1.5$, and the best a linear classifier can do is $50\,\%$ test error. But for the sake of illustration of our method, we seek to learn a (parameterized) kernel $\phi$ that transforms $x$ into a space where the data is linearly separable, or at least to a space where we can achieve low misclassification loss by learning a linear classifier with logistic regression.

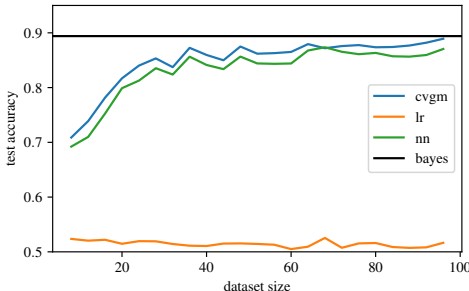

Figure 2: Test loss of the CVGM method, a neural network, and vanilla logistic regression for various dataset sizes.

We will use a one-layer neural network kernel $\phi_\alpha : \mathbf{R}^2 \to \mathbf{R}^2$, or

$$\phi_\alpha(x) = W_2\sigma(W_1 x + b_1) + b_2$$

with parameters $\alpha = [W_1, b_1, W_2, b_2]$. In our experiments, we use $\sigma(x) = \max(0, x)$, where the maximum is taken element-wise, and $W_1 \in \mathbf{R}^{64 \times 2}$ and $W_2 \in \mathbf{R}^{2 \times 64}$. We will transform the data into the new two dimensional space using the neural network, and then fit a linear classifier there using logistic regression (see Example 3.1). In other words, given $\phi_\alpha$, we minimize the following objective

$$L(\theta) = \frac{1}{2}\theta^T\theta + C\sum_{i=1}^{N} \log(\exp(-y_i v_i^T \theta) + 1).$$

where $v_i = \phi_\alpha(x_i)$. We can then find the Jacobian of the optimal solution $\theta^\star$ of this objective $\frac{\partial\theta^\star}{v_i}$ using arguments in Example 3.1, and then using the chain rule to find derivatives of the optimal solution with respect to the neural network's parameters. We use the (differentiable) soft-margin loss for the cross-validation loss function $l$, thereby allowing us to find the derivatives of the cross-validation loss with respect to the neural network parameters. A nice interpretation is that we are learning a two-layer neural network that first is fed to $\phi_\alpha$ and then to the logistic regression layer, but the first part of the neural network is trained using the CVGM, and the second is learned through (standard) logistic regression. The (differentiable) logistic regression layer is implemented as a modular PyTorch `Function`, and is in the source code. In our experiments, we fix $C = 10$, $K = 256$, $p = .95$, and use a gradient descent step size of $1 \times 10^{-1}$. For the rest of the details of the experiments, we refer the reader to the source code.

The kernel manifold at select iterations of the CVGM is displayed in the bottom part of Figure 1. After about 10 iterations, the method is able to learn a manifold under which the data is (approximately) linearly separable and achieves a test accuracy of $86.5\%$, in comparison to the Bayes-optimal accuracy on that test set of $89.2\%$.

In a separate experiment, we compared three separate methods: CVGM method with the neural network kernel as described above, a two-layer neural network with the same architecture as the CVGM method, and logistic regression. Training was done on dataset sizes from 8 to 200 over 25 random seeds. The CVGM model was trained using gradient descent on the binary cross entropy loss with a step size of $1 \times 10^{-2}$ for 100 steps, as we found that optimizing to convergence led to severe overfitting—this overfitting is more pronounced when there is less data and is likely a consequence of the low-data regime of this experiment. The mean test accuracies of the various learning algorithms over the random seeds, as well as the Bayes accuracy, are displayed in Figure 2. CVGM outperforms the two other methods, especially in the low-data regime One important observation is that CVGM is able to stably learn 322 hyperparameters with as few as 8 data points.

## 6 CONCLUSION

By showing that we can in fact differentiate the optimal solution to most convex machine learning algorithms, we have made the cross-validation loss, which is commonly viewed as a black-box

function, a differentiable objective function. This opens up the possibility of optimizing over large hyperparameter spaces, as demonstrated by our second experiment, where we optimized 322 hyperparameters with as few as 20 training examples.

Practitioners know that of the most important parts of machine learning pipelines is feature engineering, which involves applying some function to raw data before feeding it to a machine learning algorithm. Typically, the optimal features are problem-dependent, requiring experts to spend time constructing and experimenting with hand-crafted functions. However, with the CVGM, practitioners can design differentiable parameterized feature engineering functions for their class of problems, and optimize the feature engineering pipeline directly for generalization capability using gradient descent. Hence, we believe that the CVGM we present is a step towards robust automatic feature learning, a prized goal of machine learning research.

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

# Supplementary Materials

## A    SUPPORT VECTOR MACHINE EXAMPLE

Support vector machines perform classification, where $\mathcal{X} = \mathbf{R}^n$ and $\mathcal{Y} = \{0, 1\}$. The function class is again linear, or $f(x; \theta) = \theta^T x$. The loss function used in SVMs is the hinge loss, or $c(\hat{y}, y) = \max(0, 1 - y\hat{y})$. In the $\ell_1$ and $\ell_2$-regularized SVM, we optimize

$$L(\theta) = \frac{1}{N} \sum_{i=1}^{N} \max(0, 1 - y_i \theta^T x_i) + \lambda_1 \|\theta\|_1 + \frac{1}{2}\lambda_2 \|\theta\|_2^2.$$

Introducing the vectors $\theta_p, \theta_n$ (where $\theta = \theta_p - \theta_n$), variables $t_i, i = 1, \ldots, N$, we can rewrite the problem as

$$
\begin{aligned}
\text{minimize} \quad & \frac{1}{N} \sum_{i=1}^{N} t_i + \lambda_1 \mathbf{1}^T(\theta_p + \theta_n) + \frac{1}{2}\lambda_2 \|\theta_p - \theta_n\|_2^2 \\
\text{subject to} \quad & \theta_p \succeq 0, \theta_n \succeq 0 \\
& t_i \geq 0, \ i = 1, \ldots, N \\
& 1 - y_i(\theta_p - \theta_n)^T x_i \leq t_i, \ i = 1, \ldots, N,
\end{aligned}
\tag{9}
$$

which is a QP with considerable structure. Thus the solution $\theta$ is differentiable with respect to $\lambda_1$ and $\lambda_2$, again provided $\lambda_2 > 0$. A similar method, *i.e.*, replacing $x_i$ with $\phi(x_i)$ can be used to show that kernel-based SVMs are also differentiable with respect to the kernel parameters.sectionB: Learning Loss Functions

## B    LEARNING LOSS FUNCTIONS

Much of classification can be viewed as optimizing a convex surrogate of the $0 - 1$ loss function Bartlett et al. (2006). Thus, a reasonable loss function is a convex combination of such convex surrogates. Four common loss functions are:

- hinge: $l_h(\hat{y}, y) = \max(0, 1 - y\hat{y})$.
- exponential: $l_e(\hat{y}, y) = \exp(-y\hat{y})$.
- truncated quadratic: $l_t(\hat{y}, y) = \max\{1 - y\hat{y}, 0\}^2$.
- logistic: $l_l(\hat{y}, y) = \ln(1 + \exp(-2y\hat{y}))$.

Except for the hinge loss, all of these loss functions are differentiable. Thus we can define the optimization problem

$$
\begin{aligned}
\text{minimize} \quad & \frac{1}{N} \sum_{i=1}^{N} \alpha_1 t_i + \alpha_2 l_e(\theta^T x_i, y_i) + \alpha_3 l_t(\theta^T x_i, y_i) + \alpha_4 l_l(\theta^T x_i, y_i) + R(\theta, \alpha) \\
\text{subject to} \quad & 1 - y_i \theta^T x_i \leq t_i, t_i \geq 0, \ i = 1, \ldots, N
\end{aligned}
\tag{10}
$$

that is convex and differentiable in $\alpha$. We can then run a projected gradient method over $\alpha$ in the probability simplex.

## C    CONNECTIONS TO STABILITY

Bousquet and Elisseeff Bousquet & Elisseeff (2002) introduced several mathematically precise notions of the "stability" of a learning algorithm. They consider a modified dataset, constructed by *replacing* one element:

$$S^i = \{z_1, \ldots, z_{i-1}, z_i', z_{i+1}, \ldots, z_N.\}$$

The stability of a learning algorithm $\mathcal{A}$ is then defined as

$$\Delta = \operatorname*{\mathbf{E}}_{S, z_i'} \left[ l(\mathcal{A}(S), z_i') - l(\mathcal{A}(S^i), z_i') \right].$$

Roughly, this corresponds to the difference in the expected loss between $\mathcal{A}$ not having access to $z_i'$ and the algorithm having access to $z_i'$. The main theorem of the paper relates the empirical risk to population risk

$$\mathbf{E}[R - R_{\text{emp}}] = \Delta.$$

Because our goal is to optimize $\mathbf{E}[R]$, we can achieve this by optimizing the quantity $\Delta + \mathbf{E}[R_{\text{emp}}]$, or

$$\Delta + \mathbf{E}[R_{\text{emp}}] = \mathbf{E}\left[l(\mathcal{A}(S), z_i') - l(\mathcal{A}(S^i), z_i') + l(\mathcal{A}(S), z_i')\right].$$

The last two terms cancel, leaving us with

$$\mathbf{E}[l(\mathcal{A}(S), z_i')],$$

which we approximate with (3). Hence, our CVGM algorithm is implicitly choosing the hyperparameters to optimize the stability of the learning algorithm.

# D   SUPPLEMENTARY FIGURES FOR NUMERICAL EXPERIMENTS

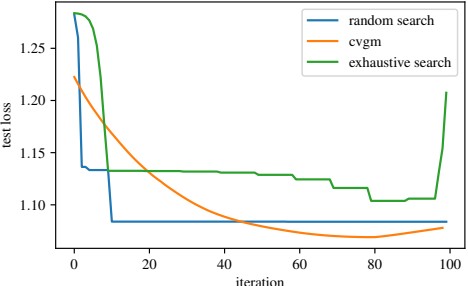

Figure 3: Synthetic regression data. Test loss of random search, CVGM, and exhaustive search, each run for 100 iterations.

# E   XOR EXPERIMENT

We also experimented with learning a two-dimensional XOR function. The data ($N = 100$) comes from two classes. One class comes from $(x, y) \sim \mathrm{U}[-3, 0.6], \mathrm{U}[-0.6, 3]$ or $(x, y) \sim \mathrm{U}[-0.6, 3], \mathrm{U}[-3, 0.6]$ with equal probability. The other class comes from $(x, y) \sim \mathrm{U}[-3, 0.6], \mathrm{U}[-3, 0.6]$ or $(x, y) \sim \mathrm{U}[-0.6, 3], \mathrm{U}[-0.6, 3]$ with equal probability. The details are similar to our classification experiment, but instead we use a two-layer neural network, with $64$ hidden units in the first layer, and $64$ hidden units in the second. (This corresponds to $4482$ hyperparameters.) The results of this experiment are displayed in Figure 4.

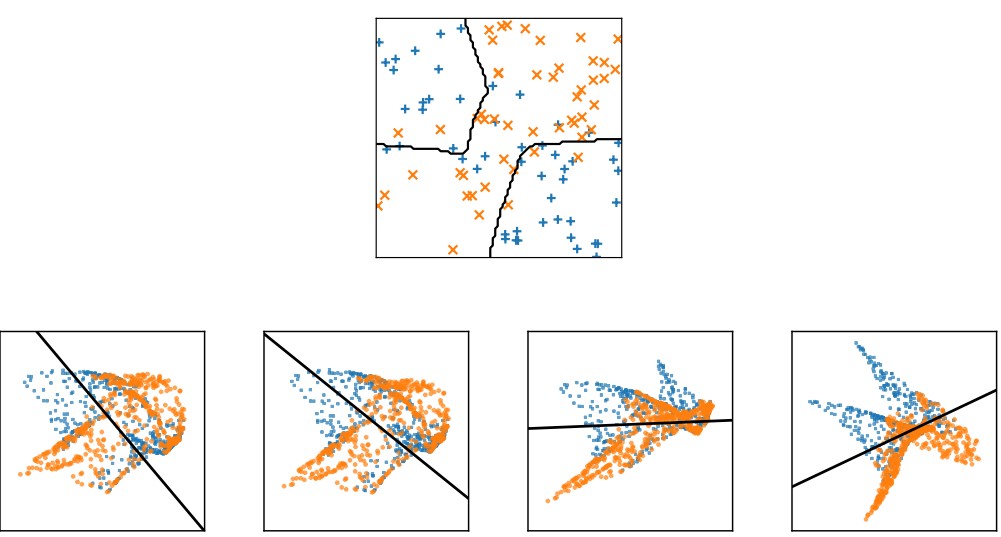

Figure 4: XOR classification experiment. Top: Synthetic training data in two dimensions, along with the learned decision boundary (black). Bottom: Visualization of kernel applied to 1000 (unseen) test data points in iterations 1, 2, 5, and 50 from left to right, along with the linear classifier in that space (black). The classifier achieves 72 % test accuracy, where the Bayes accuracy is 85.1 %.

