# OpenReview forum: "Optimizing for Generalization in Machine Learning with Cross-Validation Gradients"
_ICLR.cc/2019/Conference_

### Official Review · AnonReviewer1 · 2018-10-27
**Clearly written but incremental with respect to related work.**

**Rating:** 4
**Confidence:** 4

**Review:**

The paper proposes a method to optimize for the cross-validation performance of a model by expressing said performance as a differentiable function of the model parameters and applying a gradient-based method.

The majority of the work is clear and well-written, and appears to be correct, but I find it lacking in originality. The main contribution over related work  (references cited under the "Implicit Differentiation" heading in section 2.2) appears to be that the hyperparameters are optimized with respect to a cross-validation loss rather than a held out validation set. That is, using K=1 in the CVGM (Algorithm 1) reduces to existing work. There is also a discussion of when the cross-validation loss will be differentiable, but no new results on this. I am not sure that these contributions justify the paper.

The experiments are also not particularly strong. Only synthetic data is considered. The logistic regression baseline for the classification application in section 5.2 is irrelevant, and the neural network baseline could be clarified. Does the baseline also use the optimal parameters in the last layer throughout training? If not, how much of the improvement of the CVGM over the baseline is due to this change?

To say that the CVGM is able to stably learn the “hyperparameters” of the network kernel in this setting seems like an exaggeration -- the neural network baseline also learns these “hyperparameters”. The difference is that they are optimized with respect to the CV loss rather than the training loss.

---

### Official Review · AnonReviewer3 · 2018-10-29
**I think there is not enough novelty in this work to be considered for this conference.**

**Rating:** 2
**Confidence:** 4

**Review:**

The paper considers the problem of automatic tuning of hyperparameters in machine learning models. To address this problem the authors propose to use the so called cross-validation gradients, which optimize a validation objective with respect to the hyperparameters of a model. This approach and the investigated setting falls into a class of optimization problems known as bilevel optimization. The main characteristic of this class of optimization problems is the nested structure, with an outer and inner optimization objectives/problems. The outer problem corresponds to the validation objective and it is defined via an optimal solution to the inner problem which corresponds to a training objective. The paper, however, fails to make a reference to a rather rich literature on bilevel optimization (e.g., see [3-5] and references therein).

The approach, presented as Algorithm 1, does not seem different from [1] and [2] where hyperparameter optimization was considered for (kernel) support vector machines and (kernel) ridge regression. The data is initially split into k-folds (not necessarily of identical size) and each fold is used exactly once to define a validation objective whereas the complementary folds act as training data. The validation gradient is obtained by averaging the gradients of the k validation folds. Essentially, the same algorithm with k-fold cross-validation was considered in [1]. Thus, for me there does not seem to be any novelty in this approach and the paper itself.

The experiments involve synthetic regression and classification datasets but there are no novel insights that advance what is already known about the hyperparameter optimization (e.g., see [3]). For example, there is no intuition on the geometry of the optimization problem and the optimality of the outer optimization problem which is non-convex (e.g., see [7]), or dependence of the outer solution on the accuracy of the inner solution.

References:

[1] S. Keerthi, V. Sindhwani, and O. Chapelle. An Efficient Method for Gradient-Based Adaptation of Hyperparameters. NIPS 2007.
[2] O. Chapelle, V. Vapnik, O. Bousquet, and S. Mukherjee. Choosing Multiple Parameters for Support Vector Machines. Machine Learning, 2002.

[-3] L. Franceschi, P. Frasconi, S. Salzo, R. Grazzi, and M. Pontil. Bilevel Programming for Hyperparameter Optimization and Meta-Learning. ICML 2018.
[-4] G. Kunapuli, K.P. Bennet, J. Hu, and J-S. Pang. Bilevel Model Selection for SVMs. American Mathematical Society, 2008.
[-5] E.S.H. Neto and A.R. de Pierro. On Perturbed Steepest Descent Methods with Inexact Line Search for Bilevel Convex Optimization. Journal of Mathematical Programming and Operations Research, 2011.
[-6] B. Colson, P. Marcotte, and G. Savard. A Trust-Region Method for Nonlinear Bilevel Programming: Algorithms and Computational Experience. Computational Optimization and Applications, 2005.
[-7] M. Janzamin, H. Sedghi, and A. Anandkumar. Beating the Perils of Non-Convexity: Guaranteed Training of Neural Networks using Tensor Method. arXiv preprint arXiv:1506.08473v3, 2016.

---

### Official Review · AnonReviewer2 · 2018-10-29
**Interesting paper but with limited novelty and lacking convincing experiments**

**Rating:** 5
**Confidence:** 5

**Review:**

This paper proposes the so-called cross-validation gradient method (CVGM).
This is idea is to express the CV score as a differentiable function
of the hyperparameters and then to update hyperparameters with gradient
descent. Derivations are provided with Logistic regression and Elastic-Net
thanks to the sign splitting trick.

Once the problem is expressed as a QP, the work is mostly done
by the qpth library that offers a differentiable layer for the QP solver.

Major points:

- This idea has been around for quite some time but yes it is now certainly more
timely with the new DL tools such as pytorch. However the novelty is limited
which means that numerical experiments should be quite extensive to
demonstrate a clear impact on the field. Unfortunately the experiments
are very limited: data mostly simulated and very small. What is missing
is a real evaluation of larger datasets and a demonstration that one can
outperform the state of the art using CVGM. For example for the Elastic-Net
it is unclear if CVGM is faster than glmnet that computes full grid search
but uses warm start so is very efficient.

- Given a new dataset, how do you set step sizes? The purpose is to
find faster good hyperparameters than using Bayes Opt or random search
but if I need to fiddle with the choice of step size is it really worth it?

Minor points:

Please proof read manuscript as there are a few typos.

---

### Public Comment · (anonymous) · 2018-10-08
**A comment**

Cross-validation risk is the best surrogate of the true generalization, which is for assessment instead of being optimized however. With a strengthened optimization, cross-validation no longer reflects the generalization. The authors need to be aware of this.

---

### Meta-Review · Area_Chair1 · 2018-12-10
**Fine idea but incremental and limited experiments**

**Confidence:** 4
**Recommendation:** Reject

**Metareview:**

This paper gives explicit hyperparameter gradients for several models with convex losses.  The idea is well-motivated and clearly presented, but because it's relatively incremental, it needs a more systematic experimental section, or at least a stronger characterization of its scope and limitations.  I would also recommend an investigation of more expressive hyperparameterizations (like in Maclaurin et al 2015) and/or an investigation of overfitting on the validation set.